# Estimating the Effectiveness of Shielding during Pregnancy against SARS-CoV-2 in New York City during the First Year of the COVID-19 Pandemic

**DOI:** 10.3390/v14112408

**Published:** 2022-10-30

**Authors:** Siyu Chen, Elisabeth A. Murphy, Angeline G. Pendergrass, Ashley C. Sukhu, Dorothy Eng, Magdalena Jurkiewicz, Iman Mohammed, Sophie Rand, Lisa J. White, Nathaniel Hupert, Yawei J. Yang

**Affiliations:** 1Big Data Institute, Li Ka Shing Centre for Health Information and Discovery, Nuffield Department of Medicine, University of Oxford, Oxford OX1 3QD, UK; 2Department of Pathology and Laboratory Medicine, Weill Cornell Medicine, New York, NY 10065, USA; 3Department of Earth and Atmospheric Sciences, Cornell University, Ithaca, NY 14853, USA; 4New York Presbyterian-Weill Cornell Medicine, New York, NY 10065, USA; 5Department of Obstetrics and Gynecology, Weill Cornell Medicine, New York, NY 10065, USA; 6Department of Population Health Sciences, Weill Cornell Medicine, New York, NY 10065, USA; 7Department of Medicine, Weill Cornell Medicine, New York, NY 10065, USA

**Keywords:** pregnant women, shielding during pregnancy, effectiveness, SARS-CoV-2, New York City, dynamic model, Bayesian inference

## Abstract

Pregnant patients have increased morbidity and mortality in the setting of SARS-CoV-2 infection. The exposure of pregnant patients in New York City to SARS-CoV-2 is not well understood due to early lack of access to testing and the presence of asymptomatic COVID-19 infections. Before the availability of vaccinations, preventative (shielding) measures, including but not limited to wearing a mask and quarantining at home to limit contact, were recommended for pregnant patients. Using universal testing data from 2196 patients who gave birth from April through December 2020 from one institution in New York City, and in comparison, with infection data of the general population in New York City, we estimated the exposure and real-world effectiveness of shielding in pregnant patients. Our Bayesian model shows that patients already pregnant at the onset of the pandemic had a 50% decrease in exposure compared to those who became pregnant after the onset of the pandemic and to the general population.

## 1. Introduction

Pregnant patients make up a vulnerable patient population in any infectious disease outbreak. When New York City became the epicenter of COVID-19 pandemic in March 2020, the impact of SARS-CoV-2 infection on pregnant patients and their neonates was not well understood [1]. In addition, the prevalence of the disease in the pregnant population was difficult to capture given the lack of early testing and the presence of asymptomatic infected patients [1,2].

During this period of uncertainty early in the pandemic, most national and regional public health authorities and medical care professionals advocated for the enforcement of protective measures including wearing masks, quarantining at home when possible, and keeping social distancing. These non-pharmaceutical interventions or shielding measures have been shown to be highly effective in mitigating epidemic curves in the larger population especially during different “lockdown” periods in myriad countries [3,4,5,6] but the effectiveness among pregnant patients at that time are still unknown.

Studies have shown that pregnant patients are at higher risk of getting seriously ill from SARS-CoV-2 compared to non-pregnant patients [7,8]. A meta-analysis showed that compared to non-pregnant patients of reproductive age with COVID-19, pregnant patients are at increased risk of severe disease from COVID-19, with increased risk of ICU admission mechanical ventilation, and death [9,10]. In this study we aim to model exposure rates in the pregnant vs. general populations and evaluate the efficacy of both shielding and behavior changes during pregnancy on reducing both infection exposure and its ramifications for morbidity and mortality to SARS-CoV-2 among pregnant patients. The estimation of effectiveness of shielding during pregnancy relies on the comparison of estimates of past exposure to infection between pregnant patients and the general population. Serology tests can identify past infections and enable estimation of the number of total infections. However, naturally formed immunoglobulins targeting the virus (i.e., those generated by native infection and not vaccination) have been reported to wane below the detectable level of serological assays quite rapidly (e.g., after several months) [11,12]. The cumulative level of exposure to SARS-CoV-2 in a population therefore is not directly measurable and has to be inferred through modeling. Here, we propose a new method to estimate the cumulative exposure of SARS-CoV-2 among pregnant patients and employ a peer-reviewed model to estimate the cumulative exposure among general population in New York City, accounting for expected levels of antibody waning (seroreversion). These results have implications on future infectious disease prevention strategies in pregnancy.

## 2. Materials and Methods

### 2.1. Pregnant Patients Data

Pregnant patients giving birth at a single New York City hospital between 20 April 2020 and 27 December 2020 were included in this study. 2682 pregnant patients with clinical data capture and sample capture could have had either RT-PCR testing or serology testing or were untested (unknown). Among these 2682 patients in terms of RT-PCR, 97.7% were tested and 2.3% were unknown; in terms of serology tests, 89.9% were tested and 10.1% were unknown. For testing results breakdown: 10% were RT-PCR negative but serology unknown; 0.3% were both RT-PCR and serology unknown; 8.1% were RT-PCR negative and serology positive, 0.075% were RT-PCR unknow and serology positive, 2.1% were both RT-PCR and serology positive, 77.18% were both RT-PCR and serology negative, 2.0% RT-PCR unknown and serology negative, and 0.56% were RT-PCR positive and serology negative. The demographics of these 2682 pregnant patients can be found in Table 1. After screening the distribution of unknown tests results for PCR and serology on the calendar week, 2196 pregnant patients were included in the mathematical modelling.

The serology was detected in the serum or plasma from peripheral blood collected during admission for delivery. The serology test was performed using the clinical testing Pylon 3D platform (ET HealthCare, Palo Alto, CA). The Pylon 3D platform [13] utilizes a fluorescence-based reporting system that allows for the semiquantitative detection of antie-SARS-CoV-2 IgG and IgM with a specificity of 98.8% and 99.4%, respectively. In this paper, we denoted the serology status of every pregnant patient as positive if either IgG or IgM was positive and as negative if both IgG and IgM were negative.

Pregnant patients underwent RT-PCR testing for SARS-CoV-2 using nasopharyngeal swabs.

The observed cross-sectional data for pregnant patients is restructured into four trajectories for model fitting: weekly proportion of RT-PCR and serology negative time-series, weekly proportion of RT-PCR positive and serology negative time-series, weekly proportion of RT-PCR positive and serology positive time-series and weekly proportion of RT-PCR negative and serology positive time-series.

### 2.2. General Population Data

The seroprevalence data for general population in New York City Metro Area (including Manhattan, Bronx, Queens, Kings and Nassau) from February 2020 to December 2020 and the daily total (including confirmed and probability) mortality data were extracted from US Department of Health and Human Services Centers for Disease Control and Prevention CDC Data Tracker [14]. Details of the seroprevalence data used here can be found elsewhere [12,15].

### 2.3. Exposure Inference in Pregnant Patients

We first develop a dynamic model diagramed in Figure 1 for the temporary changing status of RT-PCR and serology among pregnant patients based on the COVID-19 disease progression. Transmission parameters specific to pregnant patients are defined in the dynamic model (Table 3). A set of ordinary differential equations (ODEs) describing the time evolution of $X_{00},\;X_{10},\;X_{11},X_{01}$ and $Z_{00}$ can be written as follows:(1)$$\begin{array}{c} \frac{dX_{00}\left(t\right)}{dt}=-\lambda_{\tau }\left(t\right)X_{00}\\ \frac{dX_{10}\left(t\right)}{dt}=\lambda_{\tau }\left(t\right)X_{00}-\tau X_{10}\\ \frac{dX_{11}\left(t\right)}{dt}=\tau X_{10}-\sigma X_{11}\\ \frac{dX_{01}\left(t\right)}{dt}=\sigma X_{11}-\beta X_{01}\\ \frac{dZ_{00}\left(t\right)}{dt}=\beta X_{01} \end{array}$$

The initial conditions of $X_{00},\;X_{10},\;X_{11},\;X_{01}$ and $Z_{00}$ at t=0 are denoted as $y_{00},\;x_{10},\;x_{11},\;x_{01}$ and $z_{00}$. Here, t=0 refers to 20 April 2020 (calendar week 17 in 2020) when the first data of pregnant patients was collected. The minimum time step in the ODEs is one week. We reparametrize the initial conditions as follows
(2)$$\begin{array}{c} x_{10}=k_{10}\left(1-y_{00}\right)\\ x_{11}=k_{11}\left(1-y_{00}-k_{10}\left(1-y_{00}\right)\right)\\ x_{01}=k_{01}\left(1-y_{00}-k_{10}\left(1-y_{00}\right)-k_{11}\left(1-y_{00}-k_{10}\left(1-y_{00}\right)\right)\right)\\ z_{00}=1-x_{10}-x_{11}-x_{01} \end{array}$$
where $\left\{k_{10},k_{11},k_{01}\right\}$ are tool parameters and constrained between 0 and 1 so that $\left\{x_{01},x_{11},x_{10},z_{00}\right\}$ can be constrained between 0 and 1. This is mainly for the convenience of MCMC implementation in Rstan. The posterior estimates of $\left\{k_{10},k_{11},k_{01}\right\}$ in each model can be found in Figure 4.

In Equation (1), $\left\{\lambda_{\tau }\left(t\right)\right\}$ is the force of infection. We first assume $\lambda_{11}\left(t\right)$ is constant over time (17≤t≤53) in Model 1 and then relax it by assuming a piece-wise constant at a fixed time step. To test the sensitivity, we try several different steps including 18 weeks in Model 2,
(3)$$\{\begin{array}{c} \lambda_{21},\;\;17\leq t<35\\ \lambda_{22},\;\;35\leq t\leq 53 \end{array}$$
12 weeks in Model 3,
(4)$$\{\begin{array}{c} \lambda_{31},\;\;17\leq t<29\\ \lambda_{32},\;\;29\leq t<41\\ \lambda_{33},\;\;41\leq t\leq 53 \end{array}$$
and 9 weeks in Model 4,
(5)$$\{\begin{array}{c} \lambda_{41},\;\;17\leq t<26\\ \lambda_{42},\;\;26\leq t<35\\ \lambda_{43},\;\;35\leq t<44\\ \lambda_{44},\;\;44\leq t\leq 53 \end{array}$$
and then compare main model results. We denote the numerical solutions of ODE system defined in (1) as ${\hat{X}}_{00},{\hat{X}}_{10},{\hat{X}}_{11},{\hat{X}}_{01}$ and ${\hat{Z}}_{00}$.

Following the dynamic model, we develop a Bayesian measurement model to model the data observation process so that the parameter estimation and model fitting can be conducted simultaneously using MCMC in Rstan [16]. The model with associated parameters (Appendix A) is described as follows:(6)$$\begin{array}{c} \lambda_{ij}\sim uniform\left(0,1\right),\;\lambda_{ij}\in \left[0,1\right]\\ \sigma \sim uniform\left(0,1\right),\sigma \in \left[0,1\right]\\ \tau \sim gamma\left(4,3\right),\tau \in \left[0.5\right]\\ \beta \sim uniform\left(0,1\right),\beta \in \left[0,1\right]\\ y_{00}\sim beta\left(8,2\right),y_{00}\in \left[0,1\right] \end{array}$$
(7)$$\left(x_{00}^{obs}\left(t\right),\;x_{10}^{obs}\left(t\right),\;x_{11}^{obs}\left(t\right),\;x_{01}^{obs}\left(t\right)\right)\;\sim Multinomial\left(N\left(t\right),{\hat{X}}_{00}\left(t\right)+{\hat{Z}}_{00}\left(t\right),{\hat{X}}_{10}\left(t\right),{\hat{X}}_{11}\left(t\right),{\hat{X}}_{01}\left(t\right)\right)$$
where $x_{00}^{obs}\left(t\right),\;x_{10}^{obs}\left(t\right),\;x_{11}^{obs}\left(t\right)$ and $x_{01}^{obs}\left(t\right)$ are the measured numbers of pregnant patients at calendar week t who were (a) both RT-PCR and serology negative, (b) RT-PCR positive and serology negative, (c) both RT-PCR and serology positive and (d) RT-PCR negative and serology positive respectively. ${\hat{X}}_{10}\left(t\right),{\hat{X}}_{11}\left(t\right),{\hat{X}}_{01}\left(t\right)$ are ODE-predicted individuals at calendar week t who were in RT-PCR positive and serology negative, RT-PCR positive and serology positive, RT-PCR negative and serology positive respectively. ${\hat{X}}_{00}\left(t\right)+{\hat{Z}}_{00}\left(t\right)$ is the ODE-predicted total number of pregnant patients at calendar week t who were either both RT-PCR and serology negative.

We use Bayesian inference (Hamiltonian Monte Carlo algorithm) in RStan to fit the model to RT-PCR and serology data by running four chains of 20,000 iterations each (burn-in of 10,000). We use 5% and 95% percentiles from the resulting posterior distributions for 90% CrI for the parameters. The Gelman–Rubin diagnostics ($\hat{R}$) given in Appendix A show values of 1, indicating that there is no evidence of non-convergence for either model formulation. Furthermore, the effective sample sizes ($n_{eff}$) in Appendix A are all more than 5000, meaning that there are many samples in the posterior that can be considered independent draws.

### 2.4. Exposure Inference in General Population

For general population in New York City, we collected morality and seroprevalence time-series data as described in the Data Description section and fitted a published model under the assumption of constant infection fatality ratio [11]. In the meanwhile, we got the estimates of cumulative exposure over time and two parameters related to the general population of New York City: they are infection fatality ratio, α and antibody decaying ratio, ω (Appendix A). Through comparing the exposure level to SARS-CoV-2 among pregnant patients and general population, we estimated the effectiveness of shielding during pregnancy.

## 3. Results

### 3.1. Dynamic Model of SARS-CoV-2 Infection

The time course of SARS-CoV-2 infection among pregnant patients can be reconstructed utilizing both RT-PCR and serology testing results by following the timeline of a typical SARS-CoV-2 infection. Most individuals, once infected, experience an incubation period before developing some symptoms of COVID-19 infection, while some individuals will remain asymptomatic throughout. The onset of RT-PCR positivity varies across individuals and types of clinical specimens [17] but systematic review studies showed that the highest percentage virus detection was from nasopharyngeal sampling between 0 and 4 days post-symptom onset at 89% (95% confidence interval (CI) 83% to 93%) dropping to 54% (95% CI 47 to 61) after 10 to 14 days [18]. In addition to testing SARS-CoV-2 RNA load using RT-PCR testing SARS-CoV-2-specific IgM and IgG antibody (in the absence of vaccination) is another method for identifying history of infection. Although the precise timing of IgM and IgG antibody detectability depends on the testing kits and varies across different individuals [19,20], on average the viral RNA is detectable one or two weeks earlier by RT-PCR than the antibody detectable by serological assays [20,21].

Assuming that the RT-PCR is positive before serology positivity, we divided the population of pregnant patients into five compartments: (1) RT-PCR negative and serology negative without previous exposure ($X_{00}$, naïve); (2) RT-PCR positive and serology negative ($X_{10}$, early phase infected); (3) RT-PCR positive and serology positive ($X_{11}$, middle-phase infected); (4) RT-PCR negative and serology positive ($X_{01}$, late-phase infected), and (5) both RT-PCR and serology negative with history of previous infection ($Z_{00},\;$past infected) (Table 2).

We next defined four transmission quantities or parameters to link these above mentioned 5 time-based compartments: force of infection, $\lambda_{\tau }$; average time lag between virus detectability by the RT-PCR test and antibody detectability by the serology assay, 1/τ; average time lag between antibody detectability by the serology assay and virus undetectability by the RT-PCR assay, 1/σ; and antibody decay rate, β (Figure 1).

The whole length of infectious period for pregnant patients can be therefore approximated by the sum of time delay between virus detectability and antibody detectability and the average time lag between antibody detectability and virus undetectability. We developed a dynamic model to study temporal changes of both RT-PCR and serology status in pregnant patients (Figure 1) with associated variables (Table 2) and parameters. Further details about the model can be found in the Methodology Section.

### 3.2. Longitudinal Cross-Sectional RT-PCR and Serology Results

We modeled the exposure of 2196 pregnant patients who delivered at a New York City hospital from 20 April 2020 through 27 December 2020 based on SARS-CoV-2 testing performed on discarded samples obtained from birth admission using data from quantitative real-time polymerase chain reaction (RT-PCR) testing for SARS-CoV-2 viral infection, or serology studies assaying levels of Immunoglobulin (Ig)G and IgM as a marker of the immune response to SARS-CoV-2 infection. Of the 2196 patients that had both RT-PCR and serology results available, 2.7% were positive and 97.3% were negative for RT-PCR testing results; and 11.2% were positive and 88.8% were negative and for serology testing results. For both tests combined, 2.2% were both positive for RT-PCR and serology, 0.5% were RT-PCR positive and serology negative, 9.0% were RT-PCR negative and serology positive, and 88.3% were both RT-PCR negative and serology negative.

### 3.3. Fitting Data from Pregnant Patients to the Dynamic Model

The test results of RT-PCR and serology allow us to divide our population of pregnant patients into four data-driven categories: (a) both RT-PCR negative and serology negative; (b) RT-PCR positive and serology negative; (c) both RT-PCR positive and serology positive; and d) RT-PCR negative and serology positive. The challenge in getting from test results to dynamic model compartments is that the first compartment ($X_{00}$, naïve) and the last compartment ($Z_{00}$, past infected) in the dynamic model (Figure 1) both manifest as both RT-PCR and serology negative, and are thus indistinguishable. To overcome this challenge, we developed a Bayesian measurement model to fit the test result data, which connects model predictions of the five time-based modeling-compartments to the measurements of the four data-driven categories.

Different models (1–4) were used to analyze different assumptions about the force infection among pregnant patients. In model 1, we assumed the force of infection is constant over time, and then relaxed the assumptions by assuming a time-varying force of infection in models 2–4 (for details of how these models differ, see the parameters in Table 3). Model fitting results showed that predictions from all four dynamic models have good agreement with measurements from the data-driven categories each calendar week (Figure 2).

### 3.4. Transmission Parameters of COVID-19 in Pregnant Patients Are Estimated to Be Consistent with Those Estimated for General Population

Data fitting allowed for the estimation of the transmission parameters. The posterior estimates of parameters for pregnant patients from the four models were summarized in Table 3. The model also estimated the proportion of patients who were giving birth but not exposed to SARS-CoV-2 ($y_{00}$) by the beginning of our study in April 2020.

We found that the estimates of the time difference between RT-PCR positivity and serology positivity, and the duration of the infectious period for pregnant patients are very robust, on average 5.5 days (95% Credible Interval, CrI (3.3, 16.7) days), and 18.8 days (95% CrI (11.3, 34.3) days), respectively. These estimates are largely comparable with those for the general population [17,22,23,24,25,26]. After seroconverting, seropositivity is estimated to be maintained for 124 days on average (95% CrI: (63, 320) days) among exposed pregnant patients. This relatively rapid seroreversion is consistent with estimates from the corresponding observational study, where analysis of the relationship between the elapsed time from the date of symptom onset and the antibody levels for pregnant patients demonstrated that the IgG positivity status could last approximately 110 days on average with a lower bound of the 95% confidence interval of 82 days but with an upper bound that is uncertain and possibly very large [2].

### 3.5. Estimated SARS-CoV-2 Exposure in Pregnant Patients Is Higher than Seropositivity Rates Would Suggest

The estimated seroprevalence (proportion of pregnant patients who are seropositive) from each of the dynamic models (Figure 3) match that of our data (Figure 2B–D). We next estimated the exposure to SARS-CoV-2 in the pregnant patients and found that exposure is estimated to be much higher than serology positivity (Figure 3). Due to the rapid decline in antibody levels after natural infection confirmed in both experimental analyses [27,28,29] and modelling analyses [11,12], there is a gap between seropositivity and the cumulative level of exposure; furthermore, this gap increases with time due to increasing exposure levels over time (Figure 3).

### 3.6. SARS-CoV-2 Exposure in Pregnant Patients at the Time of Birth Rose from Half That of the General Population to Equal That of the General Population by Late 2020

We next compared cumulative level of exposure among pregnant patients with of the general population of New York City from the same time period. In brief, the levels of exposure in general population were estimated by applying our previously published inference methodology [11] to the epidemic data including mortality and seroprevalence in general population of New York City (model fitting and parameter estimation results for the general population can be found in Appendix A and Appendix A respectively). The level of exposure in pregnant patients during April and May of 2020 is estimated to be around half of that in December 2020 in all four models (Figure 3 and Figure 4). This means that the exposure estimates of pregnant patients approaches that of the general population by November and December of 2020 (Figure 4).

Our model was structured to recapitulate the average course of SARS-CoV-2 infection with turning RT-PCR positive occurring before becoming serology positive. However, not all disease courses follow this linear model structure. It is also possible that the state of pregnancy may alter the susceptibility to SARS-CoV-2 infection, although current evidence does not support that pregnancy increases the susceptibility of infection. In addition, the antibody decaying rate may differ during pregnancy. We should note that the thresholds of seropositive and seronegative assignment might vary between assays, and the performances of assays (including sensitivity and specificity) are different. Our study is not set up for longitudinal follow-up of our cohort, thus our data is not sufficient to evaluate the impact of pregnancy on the antibody decaying rate. While more detailed longitudinal serological data could be collected and modelled during pregnancy, incorporating antibody kinetics into transmission models may hinder the applicability of estimates resulting from different assays [11].

In summary, we used a novel model to evaluate SARS-CoV-2 exposure levels in different populations using seroprevalence data and RT-PCR data, comparing exposure levels in pregnant patients in New York City to the levels in the general City population. This permits us to quantify the impact of shielding measures in preventing exposure during pregnancy across the first year of the pandemic. We estimate the impact of self-protection on reducing the level of exposure among pregnant patients during early 2020 who gave birth in this New York City hospital to be approximately 50%. These results, showing time-varying differences in exposure to SARS-CoV-2 in pregnant compared to non-pregnant populations, may have led to significant reduction in maternal morbidity and mortality in the early months of the pandemic. The estimated total exposure in pregnant patients and general population of New York City are both more than double the latest serology positive measurements.

## 4. Discussion

Positive results from RT-PCR testing and serology testing can both be used to identify infected or recently infected individuals. While an infected individual turns RT-PCR positive and then RT-PCR negative within the span of days to a week, a positive serology test result can serve as a maintained marker of infection that last for months. By capturing this dynamic effect of antibody waning in our models, we found that SARS-CoV-2 exposure estimates were much higher than the seroprevalence estimates for our sample of pregnant patients and the general public in New York City. These results confirm that previous studies looking at RT-PCR positive testing rates or seroprevalence alone will substantially underestimate population-level and subgroup exposure to SARS-CoV-2.

We found that patients who gave birth between April and August of 2020 had lower levels of exposure to SARS-CoV-2 compared to the general population. In fact, in the first months of the pandemic (April and May 2020), the exposure levels of pregnant patients were half of the exposure levels of the general population in New York City, and half of the exposure levels in pregnant patients who gave birth by the end of 2020. To understand the possible variables that contribute to this lower exposure level in pregnant patients who gave birth early in 2020, we must take into account the distinctions between the experience of pregnant patients who gave birth in early 2020 vs. late 2020. Patients that gave birth before August 2020—before the level of exposure in pregnant patients became comparable to that of non-pregnant patients—were all at least in their mid to late first trimester by the time that the pandemic hit New York City. This means that most of these patients had a high probability of knowing about their pregnancy at the onset of the pandemic, and it is possible that this knowledge of pregnancy led to behavior changes that made them more cautious than the general population. In contrast, the patients giving birth towards the end of 2020 were not pregnant and/or did not know of their pregnancy before the onset of the pandemic and may not have behaved differently than the general population; in other words, they could be considered part of general population in early 2020. During the early part of the pandemic, the population only had access to shielding measures and other non-pharmaceutical measures for prevention of disease exposures (since vaccinations only became available for the general population in early 2021). Thus, the reduction of exposure in pregnant patients by about half early in the pandemic may be attributed to effectiveness of shielding measures (Appendix A). Our current data do not address whether pregnant patients (especially those that gave birth early in the pandemic) were more stringent than the general population in following recommendations for behavioral changes and other non-pharmaceutical interventions, or whether they had additional means of improving the efficacy of shielding in preventing exposure. It is less likely that biologic differences from the state of being pregnant contributed to exposure differences as the pregnant patients that gave birth later in 2020 had similar exposures to the general population.

Such a high-level reduction of exposure might have been associated with a reduction in infection and especially a reduction of severe COVID-19 illness and, consequently, in mortality in pregnant patients. A large-scale retrospective analysis from a database that covers about 20% of the American population and includes 406 446 patients hospitalized for childbirth (6380 (1.6%) of whom had COVID-19) compared outcomes for pregnant patients with and without COVID-19 from April–November 2020 [30]. It concluded that in-hospital maternal death was rare, but rates were significantly higher for patients with COVID-19 (141/100,000 patients, 95% CI 65–268) than for patients without COVID-19 (5/100,000 patients, 95% CI 3.1–7.7). The estimate of maternal death rate is consistent with the study from the UK AAP SONPM registry, where a perinatal maternal mortality rate of 167/100,000 (for patients who have COVID-19 around the time of birth) was estimated [9,31]. Further calculation shows that the 40% to 50% reduction on exposure to SARS-CoV-2 estimated by our study might have led to the prevention of 70 (95% CI 26–134) per 100,000 maternal deaths in New York City.

After the period included in our study, additional SARS-CoV-2 preventative measures in the form of vaccinations were introduced in 2021 although strict quarantine regulations were also lifted from the city by then. Pregnant patients were not included in studies testing the safety and efficacy of COVID-19 vaccines. Studies conducted since the start of vaccination distribution including those looking at the real-word implementation of vaccination have confirmed the safety and effectiveness of vaccines specifically for pregnant patients, their placentas, and their neonates [13,32,33,34,35]. In fact, one study showed that vaccinated pregnant patients had almost 50:1 lower odds of severe COVID-19 infection [13]. Our data highlights the utility of shielding measures, and argues for an integrated intervention as suggested by CDC and NHS guidelines, which includes a combination of vaccination and shielding to reduce the morbidity and mortality of COVID-19 during pregnancy.

Our study has several important strengths, the two most important of which are robust data on a cohort of pregnant patients assessed over an extended period of time tested with both RT-PCR and serology throughout 2020, and the use of a novel model for reproducibly calculating disease exposure from testing data. While the lacuna in data capture in May and June could potentially influence the performance of parameter inference, varying model assumptions on the force of infection (as detailed in the Materials and Methods) found the estimated parameters and level of exposure to be robust and therefore clarified the likely minimal impact caused by missing data.

While the pregnant patient population is from a single NYC institution which may not be representative of the broader population, this population allowed for uniformity in testing and the study of a large cohort of patients.

## Figures and Tables

**Figure 1 viruses-14-02408-f001:**
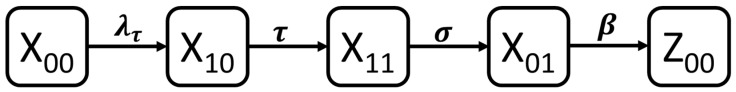
Schematic diagram of the dynamic model structure for RT-PCR and serology status.

**Figure 2 viruses-14-02408-f002:**
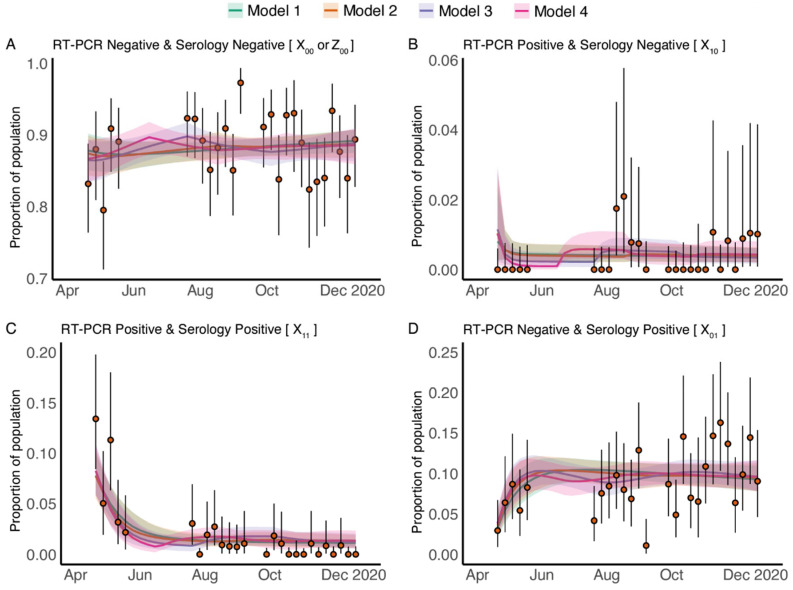
Time evolution measurements and fitted test-result model estimates of the SARS-CoV-2 RT-PCR and antibody status among patients who gave birth between 20 April 2020 and 21 December 2020. Panel (**A**–**D**), respectively, shows the model fitting results for four data-driven categories: (**A**) both RT-PCR negative and serology negative; (**B**) RT-PCR positive and serology negative; (**C**) both RT-PCR positive and serology positive; and (**D**) RT-PCR negative and serology positive. In each panel, the orange solid circles and black error bars represent the measured proportion of patients who were giving birth and in one of the four RT-PCR and serology categories and their credible intervals respectively. The green, orange, purple and pink lines in each panel show the median of estimates from Model 1–4, for proportions of patients who were giving birth in each of the four categories, while the shaded areas correspond to the 90% credible intervals. The models differ in the time-dependence of the force of infection; Model 1 assumes a constant force of infection while Models 2–4 assume time-varying force of infection.

**Figure 3 viruses-14-02408-f003:**
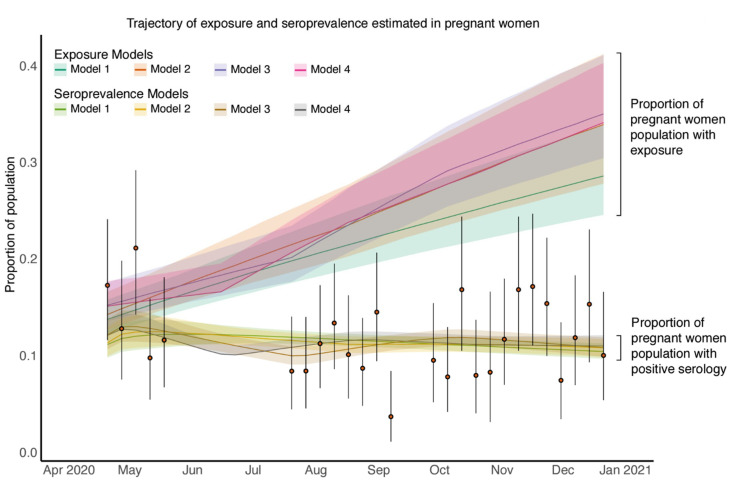
Time evolution of SARS-CoV-2 exposure and seroprevalence among patients who gave birth between 20 April 2020 and 21 December 2020. The orange solid circles and black error bars represent the measured proportion of patients who were giving birth and serology positive and their credible intervals respectively. The green, orange, purple and pink lines show the median estimates of exposure for patients who were giving birth from Model 1, Model 2, Model 3 and Model 4 respectively; shaded areas correspond to 90% credible intervals. The light green, yellow, brown and grey lines show the median estimates of seroprevalence for patients who were giving birth from Model 1, Model 2, Model 3 and Model 4 respectively; shaded areas correspond to 50% credible intervals.

**Figure 4 viruses-14-02408-f004:**
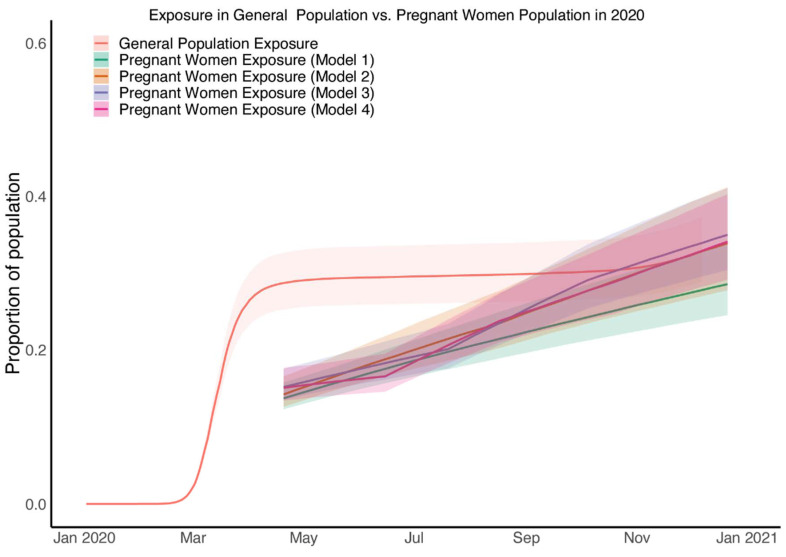
Comparisons of estimates of exposure in patients who were giving birth from four models and general population. The red line shows the median estimates of exposure from general population while the shaded areas correspond to the 95% credible interval; the green, yellow, green, blue and pink line shows the median estimates of exposure from Model 1, Model 2, Model 3 and Model 4 for patients who were giving birth, respectively, while the deep and shadow shaded areas correspond to the 50% credible intervals.

**Table 1 viruses-14-02408-t001:** Demographics table for women who giving birth prior to August 2020 and from August 2020 onwards.

	Total	Women Giving Birth Prior to August 2020	Women Giving Birth from August 2020 Onwards	Test	*p*-Value
	*n* = 2682	*n* = 1781	*n* = 901		
**Ethnicity**				Chi Square: 5.82	0.324
Not Hispanic or Latino or Spanish Origin	1769 (66%)	1173 (65.9%)	596 (66.1%)		
Hispanic or Latino or Spanish Origin	219 (8.2%)	142 (8%)	77 (8.5%)		
African American	1 (0%)	1 (0.1%)	0 (0%)		
Multi-racial	1 (0%)	0 (0%)	1 (0.1%)		
Declined	600 (22.4%)	396 (22.2%)	204 (22.6%)		
Unknown	92 (3.4%)	69 (3.9%)	23 (2.6%)		
**Race**				Chi Square: 11.49	0.244
White	1346 (50.2%)	876 (49.2%)	470 (52.2%)		
Asian	336 (12.5%)	224 (12.6%)	112 (12.4%)		
Black or African American	169 (6.3%)	118 (6.6%)	51 (5.7%)		
American Indian or Alaska Nation	6 (0.2%)	2 (0.1%)	4 (0.4%)		
Nat. Hawaiian/Oth. Pacific Island	3 (0.1%)	2 (0.1%)	1 (0.1%)		
Ashkenazi Jewish	2 (0.1%)	1 (0.1%)	1 (0.1%)		
Multiple races reported	15 (0.6%)	7 (0.4%)	8 (0.9%)		
Other combinations not described	258 (9.6%)	170 (9.5%)	88 (9.8%)		
Declined	464 (17.3%)	319 (17.9%)	145 (16.1%)		
Unknown	83 (3.1%)	62 (3.5%)	21 (2.3%)		
**Mom Age (SD) years**				*t*-test: −0.47	0.636
	34.4 (5.0)	34.4 (5.0)	34.5 (5.0)		
**Gestational Age at delivery (SD) weeks**				*t*-test: 1.53	0.126
	38.8 (2.1)	38.8 (2.0)	38.7 (2.4)		

**Table 2 viruses-14-02408-t002:** A list of patient compartments or model variables and their definitions.

Variables	Definition
$X_{00}$	proportion of naïve population who are both RT-PCR and serology negative and never exposed
$X_{10}$	proportion of early phase infected population who are RT-PCR positive but serology negative
$X_{11}$	proportion of middle-phase infected population who are both RT-PCR and serology positive
$X_{01}$	proportion of late-phase infected population who are RT-PCR negative but serology positive
$Z_{00}$	proportion of past infected population who are both RT-PCR and serology negative but previously exposed

**Table 3 viruses-14-02408-t003:** Parameter estimates (associated 90% credible intervals) among pregnant patients for each model fit.

Parameter (Unit)		Definition	Model	Median	5%	95%
$\tau^{-1}$ (days)		average time lag between virus detectability and antibody detectability	1	7	4	18
			2	5	3	16
			3	5	3	10
			4	6	4	13
$\sigma^{-1}$ (days)		average time lag between antibody detectability and virus undetectability	1	22	14	37
			2	18	11	32
			3	17	11	27
			4	18	12	28
$\beta^{-1}$ (days)		average time lag between seroconversion and seroreversion among pregnant patients	1	152	84	336
			2	118	64	270
			3	110	65	208
			4	117	66	240
$y_{00}$(−)		proportion of patients who were giving birth and not exposed by 20 April 2020	1	0.87	0.79	0.90
			2	0.86	0.76	0.90
			3	0.86	0.74	0.89
			4	0.85	0.74	0.89
$\lambda_{\tau }$	$\lambda_{11}$(−)	force of infection	1	0.0052	0.0022	0.010
	$\lambda_{21}$(−)		2	0.0063	0.0028	0.013
	$\lambda_{22}$(−)			0.0079	0.0025	0.0182
	$\lambda_{31}$(−)		3	0.0041	0.0052	0.019
	$\lambda_{32}$(−)			0.011	0.0052	0.019
	$\lambda_{33}$(−)			0.0077	0.0030	0.019
	$\lambda_{41}$(−)		4	0.00013	0.000088	0.00072
	$\lambda_{42}$(−)			0.0095	0.0051	0.016
	$\lambda_{43}$(−)			0.0070	0.0013	0.0178
	$\lambda_{44}$(−)			0.0083	0.0033	0.019

## Data Availability

All code and materials used in the analyses can be accessed at: https://github.com/SiyuChenOxf/COVID-19Exposure-ShieldingPregnantWomen (accessed on 1 October 2022). All parameter estimates and figures presented can be reproduced using the code provided. This work is licensed under a Creative Commons Attribution 4.0 International (CC BY 4.0) license, which permits unrestricted use, distribution, and reproduction in any medium, provided the original work is properly cited. The datasets from pregnant patients can be made available from the corresponding authors on reasonable request. The seroprevalence data for general population in New York City Metro Area and the daily total (including confirmed and probability) mortality data were ex-tracted from US Department of Health and Human Services Centers for Disease Control and Prevention CDC Data Tracker (https://covid.cdc.gov/covid-data-tracker/#datatracker-home, accessed on 1 October 2022).

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
