# Peer review of "Estimating the Effectiveness of Shielding during Pregnancy against SARS-CoV-2 in New York City during the First Year of the COVID-19 Pandemic"

_viruses, 2022, doi:10.3390/v14112408_

Round 1

Reviewer 1 Report

This manuscript proposed a new method to estimate the cumulative exposure of SARS-CoV-2 among pregnant patients and employ a peer-reviewed model to estimate the cumulative exposure among general population in New York City. By using the model, they found that patients already pregnant at the onset of the pandemic had a 50% decrease in exposure compared to those who became pregnant after the onset of the pandemic and to the general population, and the reduction exposure in pregnant patients by about half early in the pandemic may be attributed to effectiveness of shielding measures, such as wearing a mask, quarantining at home and keeping social distance.

Major concerns:

For these two populations, patients already pregnant at the onset of the pandemic and patients became pregnant after the onset of the pandemic, what is the characterizations of these two populations? Because it has been shown that pregnant women who were older, and those with preexisting hypertension, diabetes, asthma or cardiac disease were more at risk of severe disease. So if the patients who became pregnant after the onset of the pandemic were older and with preexisting diseases, they were more susceptible to viral infection, leading to the possible misleading conclusion. 

Minor concerns:

1. Table 1, for X10, “RT-PCT” changes to “RT-PCR”.

2. In the manuscript, fix the “Error! References source not found”.

3. It is worth mentioning in the manuscript that first SARS-CoV-2 variant have not been identified in the US during the study period (4/20/2020-12/27/2020). This is very important because the new variants are more contagious and suggested to be responsible for increase in COVID-19 infections.

Author Response

Comment 1: Define the characteristics of the pregnant population stratified by their awareness of pregnancy status during the course of 2020. It would be helpful for the authors to present the first table from the supplementary materials in the main manuscript, but to stratify it into two main groups: women giving birth prior to August 2020 and women giving birth from August 2020 onwards.

Response 1: Thank you for this valuable comment. We think it would be very helpful to define the characteristics of the pregnant population stratified by their awareness of pregnancy during the course of 2020. We adapted the advice and presented the first table from the supplementary materials in the main manuscript as Table 2 on page 3 in the revised manuscript. More than that, we presented the table by splitting the whole patient population into two main groups: women who giving birth prior to August 2020 and women giving birth from August 2020 onwards and then describing their demographics respectively. The statistical tests showed there was no significant difference in terms of demographics among these two groups.

Comment 2: Correct spelling from "RT-PCT" to "RT-PCR" on pages 2 and 11.

Response 2: Yes, we have already corrected the spelling from "RT-PCT" to "RT-PCR" at Line 84 on pages 2 and Line 400-403 on page 11 in the revised manuscript.

Comment 3: Correct the hyperlink error at line 97 on page 3.

Response 3:  Yes, we have already corrected the hyperlink error at line 96 on page 3 in the revised manuscript.

Comment 4: In the paragraph (lines 99 through 110) on page 3, add that all 2196 pregnant women had RT-PCR and serology results available.

Response 4:   Yes, we have already added ‘all 2196 pregnant women had RT-PCR and serology results available’ at Line 107 on page 3.

Comment 5: "unknow" to "unknown" in line 314.

Response 5: Yes, we have already corrected "unknow" to "unknown" at Line 320 on page 10 in the revised manuscript.

Comment 6: Correct "codes" to "code" in line 427.

Response 6: Yes, we have already corrected "codes" to "code" at line 433 on page 13 in the revised manuscript.

Reviewer 2 Report

I accept in present form

Author Response

Thank you very much for your review.